# Prader–Willi Syndrome and Chromosome 15q11.2 BP1-BP2 Region: A Review

**DOI:** 10.3390/ijms24054271

**Published:** 2023-02-21

**Authors:** Merlin G. Butler

**Affiliations:** Department of Psychiatry and Behavioral Sciences, University of Kansas Medical Center, 3901 Rainbow Blvd., MS 4015, Kansas City, MO 66160, USA; mbutler4@kumc.edu

**Keywords:** Prader–Willi syndrome (PWS), PWS molecular genetic classes, typical 15q11-q13 Type I, Type II deletions, 15q11.2 BP1-BP2 deletion, clinical findings

## Abstract

Prader–Willi syndrome (PWS) is a complex genetic disorder with three PWS molecular genetic classes and presents as severe hypotonia, failure to thrive, hypogonadism/hypogenitalism and developmental delay during infancy. Hyperphagia, obesity, learning and behavioral problems, short stature with growth and other hormone deficiencies are identified during childhood. Those with the larger 15q11-q13 Type I deletion with the absence of four non-imprinted genes (*NIPA1, NIPA2, CYFIP1, TUBGCP5*) from the 15q11.2 BP1-BP2 region are more severely affected compared with those with PWS having a smaller Type II deletion. *NIPA1* and *NIPA2* genes encode magnesium and cation transporters, supporting brain and muscle development and function, glucose and insulin metabolism and neurobehavioral outcomes. Lower magnesium levels are reported in those with Type I deletions. The *CYFIP1* gene encodes a protein associated with fragile X syndrome. The *TUBGCP5* gene is associated with attention-deficit hyperactivity disorder (ADHD) and compulsions, more commonly seen in PWS with the Type I deletion. When the 15q11.2 BP1-BP2 region alone is deleted, neurodevelopment, motor, learning and behavioral problems including seizures, ADHD, obsessive-compulsive disorder (OCD) and autism may occur with other clinical findings recognized as Burnside–Butler syndrome. The genes in the 15q11.2 BP1-BP2 region may contribute to more clinical involvement and comorbidities in those with PWS and Type I deletions.

## 1. Introduction

Prader–Willi syndrome (PWS) is caused by genomic imprinting errors with absence of expression of imprinted genes in the paternally derived PWS/Angelman syndrome (AS) region involving the chromosome 15q11.2-13 region by several genetic mechanisms. The most common cause is a paternal deletion followed by maternal disomy 15 where both 15s are from the mother, imprinting defects or chromosome 15 abnormalities (e.g., [1,2,3,4,5]). PWS affects about one in 15,000–20,000 individuals with an estimated 400,000 cases worldwide. This rare obesity-related genetic disorder has severe infantile hypotonia accompanied by poor suck with swallowing problems, sticky saliva and failure to thrive along with hypogonadism, hypogenitalism and development delay; many of these features compose the consensus diagnostic criteria triggering genetic testing for PWS [1,6,7,8]. Unique facial features are noted in PWS including bifrontal narrowing, almond-shaped eyes and a small chin with a high palate; additionally, small hands and feet with short stature due to growth and other hormone deficiencies involving the endocrine system and sex organs, pancreas, adrenal and thyroid glands occur [1,4,6,7,8,9,10,11,12,13,14,15,16,17]. Obesity, growth anomalies and hypogonadism are due to central and peripheral mechanisms involving the hypothalamus–pituitary–gonadal axis.

Nutritional phases previously described in PWS [7,18] display clinical stages of failure to thrive during infancy and excessive eating with hyperphagia in early childhood along with reduced physical activity and a lower metabolic rate leading to severe obesity, if not controlled externally [7,14]. Hyperphagia is an important health problem leading to both mortality and causes of death [19,20] with associated comorbidities typically lasting throughout life. The most common causes of death in a large survey of PWS patients studied were respiratory failure in 31%, followed by cardiac (16%), gastrointestinal (10%), infection (9%), obesity (7%) and pulmonary embolism (7%). Choking (6%) and accidents (6%) were reported more often in childhood or as young adults. The average age of death was 29.5 years [19,20]. The mortality rate for PWS is estimated at 3% per year across an age range of 0 to 47 years and 7% per year for patients aged >30 years. The most reported causes of death in children are respiratory infections and sudden deaths [21].

Mild intellectual disabilities in PWS are noted with an average IQ of 65 and about one-third having a normal IQ but often with delayed language and motor skills (e.g., [22]). Patients with PWS have unique symptoms and associated psychiatric or behavioral problems beginning in early childhood in greater than 70% of PWS patients, including emotional disturbances and obsessive-compulsive disorders, anxiety, depression, controlling and manipulative behavior, violent outbursts, stubbornness and skin picking [7,16,23]. The severity increases with age but diminishes in older patients. A high pain threshold is present along with eating nonfood or inedible items [1,6,7,11,14,16]. They also become easily frustrated with impulsivity, have a quick response to anger and lack of flexibility. Attention deficit hyperactivity with insistence to sameness is often observed in PWS and at an early age. Early diagnosis appears to lead to an improved prognosis and allows for potential treatment approaches to impact quality of life and life expectancy [24] (see Figure 1). An emerging disorder that shares genetic components with PWS is now recognized as the 15q11.2 BP1-BP2 deletion (Burnside–Butler) syndrome. The 15q11.2 BP1-BP2 region contains four genes in common with those with PWS having a typical chromosome 15q11-q13 deletion and will be discussed later in this review. Burnside–Butler syndrome is associated with motor and developmental delays, neurobehavioral problems including dyslexia, autism and psychosis with reported congenital anomalies [7,9]. Several of these findings are common in PWS, more so in those with the larger typical deletion.

## 2. Genetics of Prader–Willi Syndrome

The most recent studies using advanced genetic testing in the largest PWS cohort to date [2] showed that a 15q11-q13 paternal deletion is found in about 60% of PWS individuals, about 35% with maternal disomy 15, and the remaining individuals with imprinting defects, chromosome 15 translocations or inversions. The 15q11-13 proximal deletion breakpoint is located at two sites (i.e., breakpoint BP1 or breakpoint BP2) at the 15q11.2 band and located within either of two large duplicons predisposing for deletion hotspots at these sites (e.g., [25,26]). The larger Type I deletion at approximately 6 Mb in size involves the proximal BP1 breakpoint near the centromere and a distal 15q11-q13 breakpoint (BP3). The Type II deletion is smaller and involves proximal BP2 site located about 500 kilobases distal to breakpoint BP1 and the more distal breakpoint BP3, a breakpoint that is common in both the typical 15q11-q13 Type I or Type II deletion subtypes (e.g., [1,2,25,26], see Figure 2).

Prader–Willi syndrome is recognized as the first example of genomic imprinting in humans with dozens of genes and transcripts identified and located between chromosome 15q11-q13 breakpoints BP1 and BP3 flanked by low copy repeats prone to non-homologous recombination that leads to PWS. Genes in this 15q11-q13 region are both imprinted (*NDN*, *MAGEL2*, *MKRN3*, *SNURF-SNRPN*, *SNORDs*, *UBE3A*, *ATP10A*) and non-imprinted (*NIPA1*, *NIPA2*, *CYFIP1*, *TUBGCP5*, *GABA* receptors, *OCA2* albinism). There are 165 recognized human and 197 mouse genes currently known to be imprinted or active depending on the parent of origin. Several genes in the 15q11-q13 region have been implicated in neurodevelopment and function with a role in behavior and learning, ataxia, hyperphagia and obesity, magnesium transportation, hypogonadism and precocious puberty, circadian rhythm, autism and skin pigment production with albinism [1,2,7,17,24,25,26,27,28].

Information about the functional status of chromosome 15 genes, both imprinted and non-imprinted have been characterized. Specifically, the *NDN* (neurally differentiated EC cell-derived factor) gene which interacts with hundreds of encoded proteins such as brain-derived neurotrophic factor (BDNF) and ubiquitin E3 ligase has been studied which leads to degradation of the proapoptotic or cell cycle apoptosis regulatory protein. Additionally, *MAGEL2* or melanoma antigen-like 2 gene is imprinted in the brain and expressed from the paternal chromosome 15 allele. This gene is intron-less and associates with ubiquitin E3 ligase by altering activity, substrate specificity and subcellar location. Nonsense mutations of the *MAGEL2* gene are reported in Schaaf–Yang syndrome. At an early age, individuals with this syndrome have overlapping features including hyperphagia seen in PWS (e.g., [7,8]). The *MKRN3* or Makorin ring finger protein 3 gene is also imprinted and expressed on the paternal allele. The *MKRN3* gene plays a role in puberty and is expressed in the hypothalamus. It blocks transcription of *KISS* (KiSS-1 metastasis suppressor) and *TAC3* (tachykinin precursor 3), which are important for release of GnRH (gonadotrophin-releasing hormone) which initiates puberty [7,17]. *SNURF-SNRPN* (SNRPN upstream reading frame (SNURF)-small nuclear ribonucleoprotein polypeptide N (SNRPN)), a complex gene locus belonging to the *SNRPN* SmB/SmN family. The protein plays a role in pre-mRNA processing, tissue specific alternative splicing events and transcript production. *SNURF-SNRPN* is bi-cistronic in nature with over 100 exons that undergo alternative splicing and encodes two different proteins with exons 1–3 for SNURF producing a polypeptide and exons 4–10 generating a spliceosome protein (SmN) involved in mRNA splicing. The 5′ untranslated region component of this gene is identified as an imprinting center. This gene hosts six snoRNAs which are regulated or under the control by expression of the *SNURF-SNRPN* complex gene locus. SnoRNAs do not encode or generate protein but can impact the expression of genes and function of related proteins. Errors of paternally expressed genes/transcripts in this region do cause PWS and a second embedded imprinting center controls the maternally expressed *UBE3A* (ubiquitin-protein ligase E3A) gene causing Angelman syndrome when altered (deleted, mutated or by paternal disomy 15) (e.g., [7,16,17,29]).

*SNORD116* is one of the snoRNAs in the chromosome 15q11-q13 region and is expressed in the hypothalamic region of the brain which regulates appetite, leading to obesity [17]. It appears to play a key role in the development of features seen in PWS as recognized in those with a deletion of this transcript [7]. In mice with a deletion of this transcript, postnatal growth retardation and increased food intake are noted in research studies. These mice also show dysregulation of diurnally expressed genes such as *Mtor* and circadian *Clock*, *Cry1* and *Per2* genes [7]. Sleep disturbances are also found in individuals with PWS [1,4,7,16].

Studies on prohormone convertase (PC1) are helping to further understand the clinical phenotype in PWS based on this protein involvement in several hormonal pathways and disturbances seen in PWS. These include hyperphagic obesity, hypogonadism, short stature, growth and other hormone deficiencies, hyperghrelinemia and relative hypoinsulinemia often due to impaired prohormone processing [30]. Rare microdeletions of the *SNORD116* in humans, *Snord116* knock-out mice models and induced pluripotent stem cell-derived neurons from PWS patients support the role of disturbed prohormone convertase (PC1) activity. Burnett et al. [30] in 2016 reported reduced levels and activity of prohormone convertase PC1 in their studies. They proposed and presented data that humans and mice who are deficient in PC1 display hyperphagic-related obesity. They also have high ghrelin levels (key appetite-inducing hormone produced by the stomach), hypogonadism, and decreased growth hormone and insulin levels. This leads to short stature with reduced growth and diabetes occurring as a result of impaired prohormone processing and lack of active or functional hormones required for normal response, affecting multiple organ systems as seen in PWS. For example, POMC (pro-opiomelanocortin) is a large prohormone that requires cleavage from inactive prohormone status to smaller active peptides for normal function, including for appetite regulation and normal eating behavior which is absent in PWS. Hence, PC1, which is the protein (enzyme) encoded by the *PCSK1* (proprotein convertase, subtilisin/kexin-type 1) gene located on chromosome 5q15 and is involved in post-translational modification or change of prohormone (inactive) to individual peptides (active) status, may be influential. It is suggested that several major neuroendocrine clinical findings seen in PWS could be due to PC1 deficiency requiring more investigations with the potential to lead to therapeutic agents [17,30].

## 3. Clinical Description of Prader–Willi Syndrome and 15q11.2 BP1-BP2 Deletion

Butler et al. [31] reported PWS patients and the chromosome 15q11-q13 deletion were more affected than patients with maternal disomy 15. Distinct differences were also reported in those with the two typical 15q11-q13 deletions compared with maternal disomy 15, particularly in phenotype, learning and psychiatric/behavioral parameters [31]. Specifically, Roof et al. [32] reported those with PWS and maternal disomy 15 had higher Verbal IQ scores than those with the 15q11-q13 deletion. Furthermore, PWS individuals with the deletion had more self-injury and severe behavior with lower intellectual ability than those with maternal disomy 15 [1,7,16]. The first clinical differences in individuals with PWS were described by Butler et al. [31] in 2004 when examining Type I or Type II deletions including assessments for intellectual, adaptive and aberrant behavior assessments, reading and math skills and visual-motor integration. Generally, poorer assessment scores were found in PWS individuals with Type I deletions compared to those with the smaller Type II deletions or maternal disomy 15. The larger typical Type I deletion accounts for about 40% of the typical 15q11-q13 deletions in PWS [2,31]. Specifically, those with the larger Type I deletions had more compulsions, poorer adaptive behavior and reduced cognition than those with the smaller Type II deletions. Those with the larger deletion had more severe compulsions related to grooming and bathing and compulsions that were more disruptive to daily living [1,31,33,34,35]. Intellectual ability and academic achievement skills were analyzed, and visual processing was poorer in those with the larger deletion [36,37]. Furthermore, self-injurious behaviors were more commonly observed in those with the larger Type I deletion. Hence, individuals with PWS and the larger 15q11-q13 Type I deletion including the four non-imprinted protein coding genes (*NIPA1*, *NIPA2*, *CYFIP1*, *TUBGCP5*) in the 15q11.2 BP1-BP2 region are more clinically impaired. Those individuals with 15q11.2 BP1-BP2 deletions are missing the four genes alone and do not have PWS but have Burnside–Butler syndrome (BBS) (e.g., [27,38,39]) with developmental motor and speech delays, congenital findings, behavioral problems including autism and brain imaging abnormalities (e.g., [27]).

Individuals with the larger typical 15q11–q13 Type I deletion were found to have more severe neurodevelopmental symptoms when compared to those with PWS or Angelman syndrome, a sister genomic imprinting disorder with loss of maternally expressed genes on the chromosome 15q11-q13 region with the smaller typical Type II deletions (e.g., [30,33,34,35,40,41,42]). Bittel et al. [43] reported on molecular gene expression (messenger-RNA) studies from lymphoblastoid cells obtained from PWS males and females from highly conserved *NIPA1*, *NIPA2*, *CYFIP1* and *TUBGCP5* genes. They found that 24–99% of the phenotypic variability seen in behavioral and academic measures in PWS subjects could be explained by individual gene expression patterns. Levels of messenger-RNA from *NIPA1*, *NIPA2*, *CYFIP1* and *TUBGCP5* were reduced but detectable in individuals with PWS and the Type I deletion, supporting biallelic expression. Generally, messenger-RNA values were positively correlated with assessment measures, indicating a direct relationship between messenger-RNA levels and better assessment scores. The highest correlation was for *NIPA2*. Negative associations were found between age and behavior in the Type I deletion subtype only and implicating the four genes, specifically *CYFIP1* and *NIPA2.* Disturbed expression of *CYFIP1* is seen in other developmental disabilities including those with 15q disorders without PWS [44,45] and fragile X syndrome [46]. *NIPA1* and *NIPA2* are known to encode magnesium transporters and magnesium levels were recently reported to be lower in those with PWS and the Type I deletion compared to those with Type II deletions [28,47,48].

Clinical findings were reported in the literature from 200 patients with 15q11.2 BP1-BP2 deletion (Burnside–Butler) syndrome grouped into five categories [27]. These categories were (1) developmental (73% of cases), speech (67%) and motor delays (42%); (2) dysmorphic ear (46%) and palatal defects (46%); (3) writing (60%) and reading (57%) difficulties, memory problems (60%) and verbal IQ scores ≤ 75 (50%); (4) general behavioral problems, unspecified (55%); and (5) abnormal brain imaging including white matter disease (43%). Less often seen features were seizures/epilepsy (26%), autism spectrum disorder (ASD) (27%), attention-deficit hyperactivity disorder (ADHD) at 35% of cases and schizophrenia/paranoid psychosis (20%). Furthermore, Davis et al. [49] reported the parent of origin in dozens of families and found a maternal origin of the 15q11.2 BP1-BP2 deletion to be associated with a significantly higher risk for developmental, motor and speech delays, intellectual and learning problems, autism and behavioral/psychiatric diagnoses. Those with paternal chromosome 15q11.2 BP1-BP2 deletions were more prone to poor coordination/ataxia and congenital anomalies. The 15q11.2 BP1-BP2 region is deleted in PWS individuals having the larger Type I deletion. Butler [47] and Butler et al. [48] reported on the role of the four genes found in the 15q11.2 BP1-BP2 region involving magnesium transportation in the clinical presentation and potential treatment of those with Type I deletion. Figure 3 illustrates a frontal view of a non-dysmorphic mother and child having the 15q11.2 BP1-BP2 deletion (Burnside–Butler) syndrome.

## 4. Description, Evaluation and Gene Expression of Chromosome 15q11.2 BP1-BP2

### 4.1. Chromosome 15q11.2 BP1-BP2 Region Description

One allele is missing in each of the four genes (*NIPA1*, *NIPA2*, *CYFIP1*, *TUBGCP5*) in the 15q11.2 BP1-BP2 region due to a deletion designated as Burnside–Butler syndrome, emerging with variable clinical findings including a neurodevelopmental-autism non-dysmorphic phenotype with low penetrance. This phenotype presents with features including language and/or motor delay, cognitive impairment, aberrant behavior and autism, poor coordination with ataxia, seizures and congenital anomalies [50,51,52,53,54,55,56,57]. This small proximal 15q11.2 deletion characterized by Burnside et al. [39] in 2011 is now recognized as the most common chromosome finding in large cohorts of those presenting with neurodevelopmental problems and/or autism [50].

Ho et al. [50] in 2016 used 2.8 million DNA markers including single nucleotide polymorphic (SNP) and copy number variant (CNV) probes optimized for detection of regions of homozygosity and CNVs associated with neurodevelopmental disorders with high-resolution chromosomal microarrays. Neurodevelopmental disorders include developmental delay and intellectual disabilities with ASD, which affect up to 15% of all children. About 40% of those with ASD also have learning disabilities and approximately 30% show other comorbidities including seizures (e.g., [58,59]). Genetic testing may allow pinpointing the causes critical for clinical management and genetic counseling of at-risk family members.

Ho et al. [50] summarized results in a total of 10,351 custom microarrays performed on patients over a period of four years with a male to female ratio of 2.5:1 and a mean age of 7.0 years. This neurodevelopmental patient cohort comprised 55% of cases with a diagnosis of ASD with or without other features (ASD+ and ASD only). Neurologists were the most common referring physician group at 36%, followed by developmental pediatricians at 31%, pediatricians at 16% and medical geneticists at 14%. Psychiatrists referred only 2% of the total cases but had the highest indication of ASD at 72% with or without features. In the study, 74% of ASD cases were referred by pediatric neurologists or developmental/behavioral pediatricians. Potentially abnormal CNVs were observed in 28% of cases with an average 1.2 reportable CNVs per individual. Overall detection rate for individuals with ASD was significant at 24.4%. The detection rate for a pathogenic cause using chromosome microarray analysis varied by indication for testing, age and gender as well as specialty of the ordering physician.

The most common genetic defect identified in the combined ASD group (N = 5694 patients) was the 15q11.2 BP1-BP2 deletion, and the 22q11.2 deletion was seen most often in the non-ASD group (N = 4657). The most common cytogenetic finding seen in both the ASD+ group (N = 2844) and ASD only group (N = 2850) was also the 15q11.2 BP1-BP2 deletion. This chromosome 15 defect was the most common finding in both females and males. Of the 85 genetic findings reported by Ho et al. [50], 9% of the patients had the 15q11.2 BP1-BP2 deletion, followed by the 16p11.2 deletion at 5% and 16p11.2 duplication at 5%. Other findings included the 15q13.3 deletion, 16p13.1 duplication and *NRXN1* gene deletion all at 4%. Hence, a greater understanding of the 15q11.2 BP1-BP2 deletion may further impact the role in the context of PWS, the most classical chromosome 15 disorder in humans.

### 4.2. Chromosome 15q11.2 BP1-BP2 Genes, Functions and Pathway Analysis

To further investigate biological pathways involving the 15q11.2 BP1-BP2 region, Rafi and Butler [28] examined STRING protein–protein interactions that encompass the four 15q11.2 BP1-BP2 genes with predicted Gene Ontology (GO) functions and processes. They found a role in magnesium ion transport, regulation of cellular growth and development with production of bone morphogenetic protein (BMP) and signaling pathways, regulation of axonogenesis and axon extension, cellular growth and development, and plasma membrane bounded cell projection and mitotic spindle organization. Using searchable genomic disease websites and tabulating disease data, they found the top ten overlapping significant neurodevelopmental disorders to be Prader–Willi Syndrome (PWS); Angelman Syndrome (AS); 15q11.2 Deletion Syndrome with Attention Deficit Hyperactive Disorder and Learning Disability; Autism Spectrum Disorder (ASD); Schizophrenia; Epilepsy; Down Syndrome; Microcephaly; Developmental Disorder; and Peripheral Nervous System Disease. These were also individually associated with PWS, ASD, ataxia, intellectual disability, schizophrenia, epilepsy and Down syndrome.

Of the four non-imprinted biallelic genes and their encoded proteins, NIPA1 protein interacts with 11 other proteins, of which five (45%) are bone morphogenic protein (BMP) superfamily members, three (27%) are BMP receptors and one is the TGFB1 (9%) protein [28]. Therefore, three-fourths of NIPA1 interacting proteins are important for developmental bone morphogenesis or involved in multifunctional proteins controlling proliferation, differentiation and other cellular functions. The *NIPA2* gene and protein interact with 19 other proteins and of these three (16%) are involved with the BMP protein superfamily, three (16%) proteins interact with BMP receptors ACVR1 and TGFBR1, and six are members of the SMAD superfamily of proteins (42%). These genes are important as intracellular signal transducers and transcriptional modulators activated by TGFB [28]. *NIPA1* and *NIPA2* genes also encode magnesium transporter proteins (e.g., [28,54]). Picinelli et al. [55] reported on a small number of patients with 15q11.2 BP1–BP2 deletions or duplications and found an inverse relationship for the NIPA2 gene encoding a magnesium transporter in both central nervous system (CNS) and renal tubules to be directly associated with urinary magnesium levels. Those with this gene deletion had higher urinary magnesium and those with this gene duplicated showed lower urinary levels. PWS patients with the larger Type I deletion had lower plasma magnesium levels [48]. In addition, Xie et al. [56] reported *NIPA2* gene mutations that showed incorrect localization of the NIPA2 transporter protein in neurons. This resulted in decreased intracellular magnesium levels apparently due to reduced cross-membrane transport involving renal tubules. Furthermore, Mg^2+^ is involved in gating and activation of channels and receptors including NMDARs, playing a role in memory processing and altered neural extracellular Mg2+ concentrations caused by *NIPA2* deletions in neuronal cells.

*NIPA1* (non-imprinted in PWS/AS 1) gene defects cause autosomal dominant hereditary spastic paraplegia and postural disturbance [60,61]. *NIPA1* is known to mediate Mg^2+^ transport and is highly expressed in the brain [28]. The NIPA1 protein can also transport other divalent cations such as Fe^2+^, Sr^2+^, Ba^2+^, Mn^2+^ and Co^2+^ and possibly impacting their levels. The *NIPA2* (non-imprinted in PWS/AS 2) gene encodes a protein acting as a renal Mg^2+^ transporter. Three reported *NIPA2* mutations (p.I178F, p.N244S and p.N334_E335insD) are found in childhood absence epilepsy [54,56]. *NIPA2* gene variants and functional studies have shown decreased intracellular magnesium concentrations in neurons, suggesting lower intracellular magnesium levels may enhance N-methyl-d-aspartate receptor (NMDAR) currents impacting neuron excitability and brain function.

CYFIP1 (cytoplasmic fragile X mental retardation 1 FMR1 interacting protein 1) is reported to interact with other proteins with functions related to actin filament binding with cell-matrix adhesion, cytoskeleton organization, MAP kinase signal transduction of cell growth, survival and differentiation with stimulation of glucose uptake, intracellular protein breakdown and tissue remodeling and mediation of translational repression. These interactions may impact brain morphology associated with learning and memory impairment. The *CYFIP1* gene encodes a protein that also interacts with FMRP, the protein coded by the *FMR1* gene. It is associated with fragile X syndrome, the most common cause of intellectual disabilities and autism found in families [45,62].

The *TUBGCP5* (tubulin gamma complex associated protein 5) generates an interaction with proteins involved with mitotic spindle formation and assembly along with microtubule organization and production of centrosomal proteins. These are involved in centriole duplication and regulation during cell division. They are also associated with chorioretinopathy and microcephaly (e.g., [28,44,55]) as well as ADHD and obsessive-compulsive disorder (OCD) [28].

Seven genes interact directly with the non-imprinted 15q11.2 BP1-BP2 genes including *CFHR1* or complement factor H-related protein 1 interacting with complement regulation; *SPAST* or Spastin, an ATP-dependent microtubule-severing protein involved in movement; *SPG20* or Spartin implicated in endosomal trafficking participating in cytokinesis; *CFHR3* or complement factor H-related protein involved with complement regulation; *MNS1* or meiosis-specific nuclear structural protein 1 controlling meiotic division and germ cell differentiation; *IGFBF2* or insulin-like growth factor-binding protein 2 inhibiting IGF-mediated growth and developmental rates with *BMPR2* or bone morphogenetic protein receptor 2 [26].

### 4.3. Clinical Evaluation and Findings in 15q11.2 BP1-BP2 Deletion

Clinical and brain imaging data support brain disturbances with global morphology and subcortical volume differences in those with the 15q11.2 BP1-BP2 deletion [63]. Significantly lower nucleus accumbens volume and total surface brain area were found along with thicker cortices reported in those with the deletion when compared to individuals without the deletion, typically across the frontal lobe, anterior cingulate and precentral and postcentral gyri regions using brain magnetic resonance imaging. The investigators also measured cognitive function and found lower performance on all tasks in those with the 15q11.2 BP1-BP2 deletion and larger intracranial volume and total surface area were associated with higher performance on nearly all cognitive tasks. Generally, frontal cortex surface regions were found to be associated with task performance, particularly for fluid intelligence and trail-making tasks.

In addition, onset of adverse perinatal events and early life outcomes have been examined in pregnancies with 15q11.2 BP1-BP2 anomalies [64,65]. For example, Chu et al. [64] analyzed 1,337 pregnancies with genetic amniocentesis and found that 0.7% of cases had the 15q11.2 BP1-BP2 deletion and 0.8% had a duplication of the same region. They compared the pregnancies with normal microarray results to those with the 15q11.2 BP1-BP2 deletion and more cases with the deletion received neonatal intensive care, Apgar scores less than 7 (at 1 min) and recorded neonatal deaths. Perinatal findings noted in other studies included development delays and more infantile deaths in the deletion group, specifically related to congenital heart disease [66]. Other birth defects reported at birth in those with this deletion include congenital arthrogryposis [67] and tracheoesophageal fistula with congenital cataracts [68].

The published abnormal results associated with the 15q11.2 BP1-BP2 deletion do not fit one molecular dysfunction, but multiple altered functions of the four genes, suggesting involvement in neuro-plasticity, development and function. One important interactive gene involved in this cytogenetic region and is associated with learning and motor delays, autism and schizophrenia is *CYFIP1*, which encodes a protein involved with actin cytoskeletal dynamics. It interacts with the fragile X mental retardation protein and when disturbed causes fragile X syndrome with abnormal white matter microstructure and postnatal hippocampal neurogenesis microglial disturbances [69,70,71]. For example, Silva et al. [69] applied a brain-wide voxel-based approach and analyzed diffusion tensor imaging data from healthy individuals and those with 15q11.2 BP1-BP2 deletions or duplications. A reciprocal effect was found for 15q11.2 BP1-BP2 on white matter microstructure suggesting reciprocal chromosomal imbalances may lead to opposite changes in brain structure. For example, findings in the deletion group overlapped with more white matter differences previously reported in the fragile X syndrome suggesting common pathogenic mechanisms derived from disruptions of the cytoplasmic CYFIP1–fragile X mental retardation protein complexes. In addition, other interactive genes including solute carrier (SLC) family composed of at least 43 gene families and hundreds of transporters may impact features seen in PWS and Burnside–Butler syndrome [28,52]. This collection of gene interactions may contribute to comorbidities seen in PWS, particularly those with the larger Type I deletion including glucose and insulin metabolic alterations [48]. Identified components of the 15q11.2 BP1-BP2 phenotype and neurobiological mechanisms should stimulate more studies to test similarities between the 15q11.2 BP1-BP2 disorder and in those with PWS having the larger typical 15q11-q13 Type I deletion with these four genes deleted.

## 5. Conclusions

Bi-allelic *NIPA1*, *NIPA2*, *TUBGCP5* and *CYFIP1* genes located in the proximal 15q11 chromosomal region between breakpoints BP1 and BP2 are implicated in compulsivity, aberrant behavior and lower intellectual ability in individuals with Prader–Willi syndrome with Type I versus Type II deletions. For example, the coefficient of determination for the deletion type alone explained 5 to 50% of the variation in the clinical parameters that were assessed when examining expression of the four genes [43].

In summary, this chromosome deletion has been unequivocally associated with congenital defects, neurobehavior disturbances including autism, schizophrenia, dyslexia or dyscalculia in the vast majority of individuals studied related to these four genes. Additional studies are needed to identify the function of the four genes and their interaction with gene networks to clarify their potential role and include brain gene expression patterns with involved pathways. Parent of origin and gender-based differences should be studied. These may impact fetal phenotype, prognosis and development/behavior/psychiatry with the need for follow-up and long-term care, including echocardiography due to the higher risk of congenital anomalies and heart defects with developmental assessments from disturbances seen in those with the 15q11.2 BP1-BP2 deletion.

The clinical presentation and findings seen in Burnside–Butler syndrome (BBS) are also seen in individuals with PWS, particularly those with the larger Type I deletion coined PWS and BBS, with mounting evidence of more severity than is observed in other genetic defects in PWS such as the smaller Type II deletion or maternal disomy 15. The description and explanation of the clinical findings focused on this report and associations could be recognized as a separate category of PWS, particularly in the 25 to 30% of all PWS patients who have the larger Type I deletion with loss of the four described genes and their interactions. These observations could lead to the potential for earlier intervention, treatment and surveillance when applied at a young age, particularly magnesium surveillance and supplementation, if low, with close medical care. More research is requested to pursue and confirm these potentially associated findings with impact on prognosis, longevity and quality of life.

## Figures and Tables

**Figure 1 ijms-24-04271-f001:**
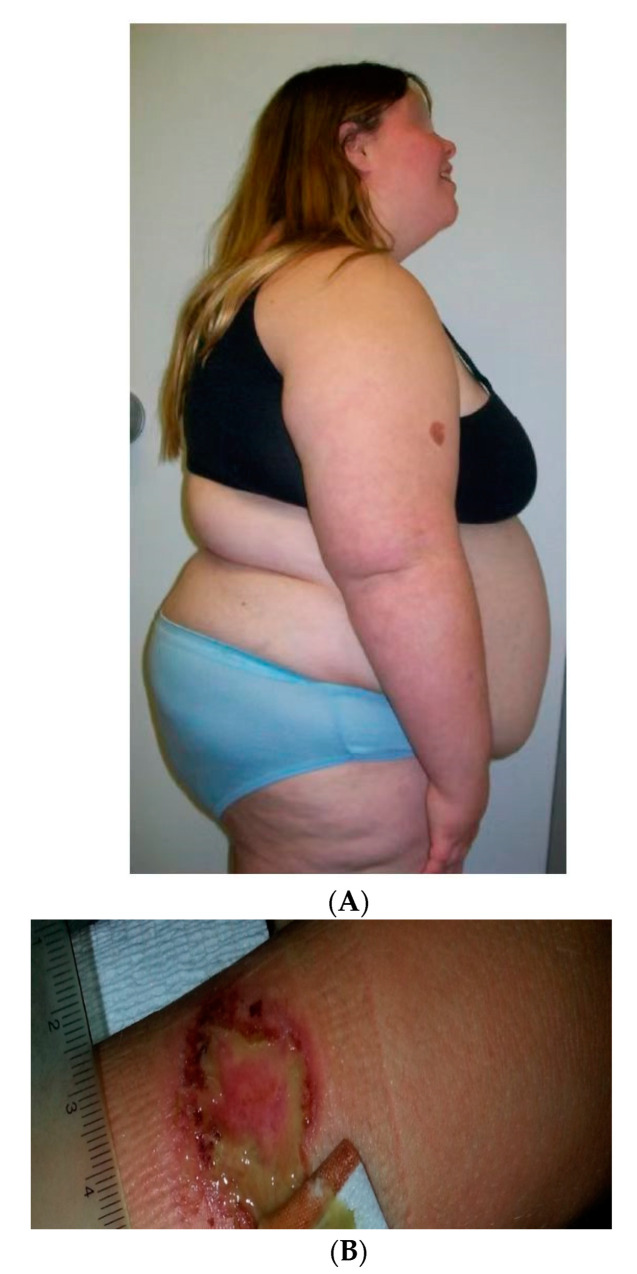
(**A**) A profile view from a 16-year-old female with Prader–Willi syndrome (PWS) showing central obesity as a major feature of this disorder. (**B**) A self-injury site noted in a separate patient with PWS, an abnormal clinical finding often seen in PWS.

**Figure 2 ijms-24-04271-f002:**
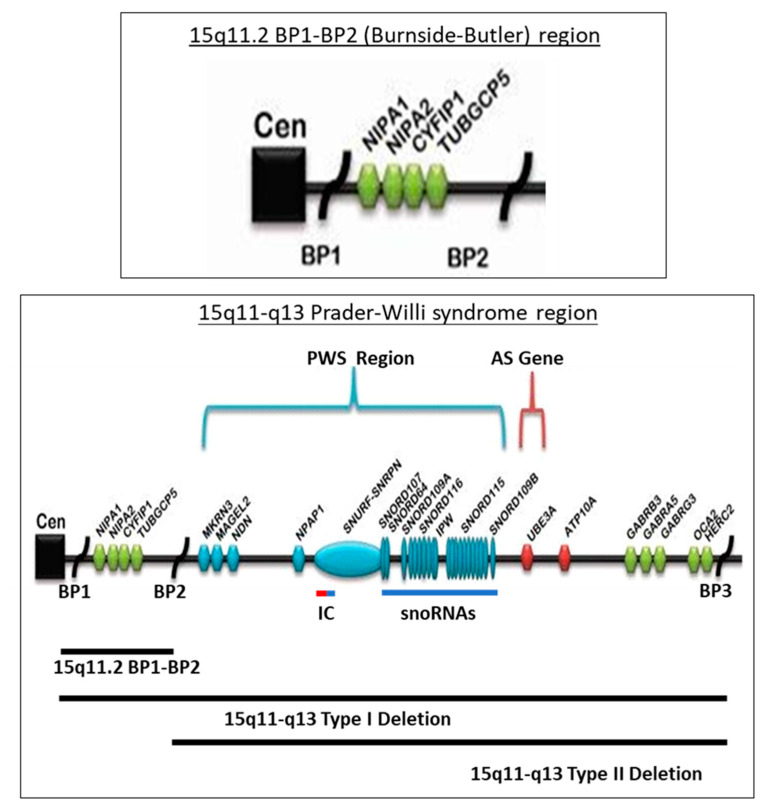
Chromosome 15q11-q13 region with gene and transcript symbols in blue and the location causing Prader–Willi syndrome (PWS) imprinted with paternal expression. Angelman syndrome (AS) and maternal expression is in red, including the causative AS gene *(UBE3A*). The non-imprinted genes are green. The three 15q11-q13 breakpoints (BP1, BP2 and BP3) are the sites for the three chromosome 15q deletions; the larger Type I at BP1 and BP3, smaller Type II at BP2 and BP3 and the 15q11.2 BP1-BP2 deletion alone designated as Burnside–Butler syndrome. IC designates the imprinting center that controls the activity of the imprinted genes in the region and dependent on the parent of origin. The 15q11.2 BP1-BP2 region is enlarged and illustrated at the top.

**Figure 3 ijms-24-04271-f003:**
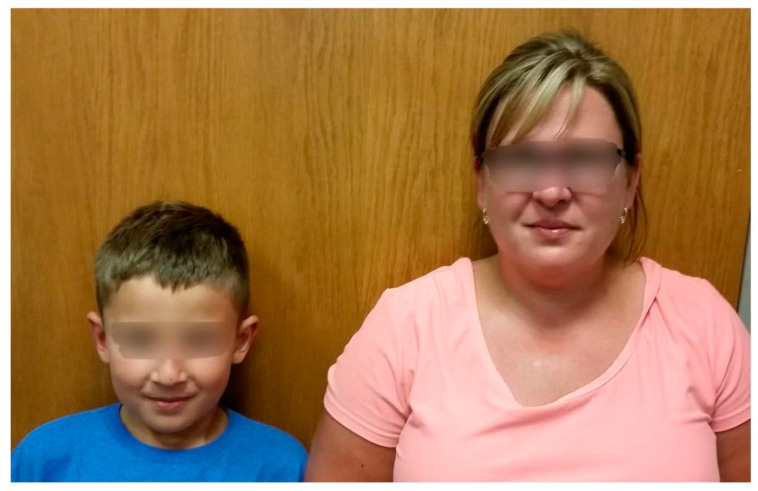
Frontal views of a mother and child with the 15q11.2 BP1-BP2 deletion (Burnside–Butler) syndrome.

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
