# Peer review of "Prader–Willi Syndrome and Chromosome 15q11.2 BP1-BP2 Region: A Review"

_ijms, 2023, doi:10.3390/ijms24054271_

Round 1
Reviewer 1 Report
Prader-Will Syndrome and Chromosome 15q11.2 BP1-BP2 Region: A Review
The author reviews the symptoms of Prader-Will syndrome, its genetic causes, and the genes located in the affected region of chromosome 15.
Comments
- Section 4 becomes difficult to follow. I would recommend structuring in subsections according to the gene, or according to the phenotype they affect.
- Names of genes must be written in italics.
- The first time a gene is mentioned, it is advisable to put the full name followed by the abbreviation or the name by which it is known in the databases in parentheses. This has not been done in all cases, and in those that have, a uniform rule has not been followed, such as putting the full name in parentheses.
- Page 6, line 209: the meaning of the abbreviation ASD is not specified the first time it is used.
- Page 7, line 263: the meaning of the abbreviation CNS is not specified the first time it is used, although it is not necessary to abbreviate something that is only used once.
Author Response
Thank you for the review and suggested changes. All of my changes/revisions are in red throughout the manuscript and will be sending the manuscript when all changes are made. I have added subsections to Section 4 as recommended. I have itialized all gene symbols as recommended at definition at the location when the gene and its function is first described as commonly done. The abbrevations were supplied throughout.
Reviewer 2 Report
The 15q11-q13 region harbors three low copy repeats of that constitute breakpoints for several neurobehavioral disorders. BP1 and BP2 are separated by about 0.5Mb, while BP3 is about 6Mb distal. The imprinting disorders Prader Willi or Angelman syndromes can result from deletions spanning from either BP1 or BP2 to BP3. In both syndromes, the larger BP1 to BP3 deletion usually presents with more severe clinical findings than the smaller BP2-BP3 deletion. Several years ago, Dr. Butler led the way in investigating BP1 to BP2 deletions which are now known as Burnside Butler syndrome. In this review Dr. Butler provides and overview clinical differences between PWS individuals with the larger BP1-BP3 deletion compared to those with the BP2-BP3 deletion. The larger deletion correlates with more compulsions, poorer adaptive behaviors, worse visual processing, and more self-injurious behavior. Clinical findings from individuals with BP1-BP2 deletions are described next, followed by a molecular genetic analysis of the four non-imprinted protein coding genes in the region, and the contribution of the region to autism. The biochemical roles of these four genes are discussed next, including an analysis of how hemizygosity for these genes may contribute to traits seen in BBS and PWS. Lastly, Dr. Butler provides a summary of the material presented and suggests needed areas of knowledge lead ing to better outcomes for affected individuals.
The review is simultaneously extensive and concise, and is nicely referenced. It reads very well and should be accessible to those with a limited knowledge of medical genetics or this busy region of the genome. A quick Pubmed survey indicates that while individual aspects of BBS have been recently reviewed, the last review of BBS clinical and genetics findings was in 2017, also by Dr. Butler.
Author Response
Thank you for the suggested changes. I have addressed the objective in comparing the features between PWS and BBS with overlap genetically with 15q11.2 BP1-BP2 deletion as a component of PWS patients with the large Type I 15q11-q13 deletion as the last few sentences of the Introduction section. I have also rewrote the description of the PWS molecular genetic subtypes in line 27-29 in original manuscript. I have added 'genetic' after 'rare' and before 'obesity' in line 31. A sentence on consensus diagnostic criteria was added but advanced genetic testing has replaced the consensus diagnostic criteria as the diagnosis of PWS is now genetically confirmed via advanced laboratory testing. Figure 1 has been altered by decreasing the size of the PWS female and adding A. and B. subgroups within Figure 1 and working altered as suggested. More detailed information was added on the 15q11-q13 region with more description on genes/transcripts and functions as also suggested by another reviewer including SNORD116. The flanked low copy repeat information at the chromosome breakpoints were added as suggested. The comment about evidence of chromosome 15 inversion in PWS has been noted previously by Winsor and Welch, 1983 (PMID: 6652960). Comments were made about modifying/editing Figure 2 and will undertake this task. Thank you
Reviewer 3 Report
duc included

Author Response
Thank you for the comments which have been addressed involving all reviewers and changes in the revised manuscript are found in red. These include addressing the objective of paper commit and revisions found in last paragraph of Introduction section. The first comment in Introduction section was addressed by rephrasing. The word 'genetic' was added to the 'rare obesity' statement as recommended. Consensus diagnostic criteria statement was added but advanced genetic testing is now used to genetically confirm and diagnose PWS. The pictures in Figure 1 were made smaller and figure subgroup developed as recommended and rephrased. A more complete description of the imprinted and non-imprinted genes in the 15q11-q13 region was made as noted in the revisions. The comment regarding flanking low copy repeats at the 15q11-q13 breakpoints was added. Parental inversions of chromosome 15 have been reported in PWS including by Winsor and Welch, 1983 (PMID: 6652960). I am in the process of editing the title and contents within the chromosome region (Figure 2) in order to address the comments. SNURF-SNRPN as a unit has been added. The comments have been addressed regarding the references and genes. Lines 150-153 have been rewritten. The comment on parent of origin was addressed as several dozen families were contacted and reviewed as one of the coauthors of the Davis et al report but would encourage more studies. The four genes in the 15q11.2 BP1-BP2 or most often involved in neurodevelopment and function as supported by structural brain differences and reported as generally non-dysmorphic externally and now noted in the text. Old Line 197 has been rewritten and low penetrance was added. Old line 237- is as a result using computational biology/analysis with STRING with pathways and disease association programs when analyzing gene-gene-protein interactions and relationship with the four genes and 15q11.2 BP1-BP2 region as reviewed by Rafi and Butler, 2020. Magnesium is key metal required for brain and muscle function and interaction; important for findings seen in BBS supported by differences in magnesium disturbances in the Type I and Type II PWS patient studies and two of the four genes encode magnesium transporters. Butler et al. have been involved with the first and most comprehensive studies and reviews undertaken and reported. He has evaluated patients as a physician scientist and has undertaken research for over two decades on this topic with 100s of publications and a key opinion leader on the topic, both PWS and BBS.
Reviewer 4 Report
Authors should be congratulated for their work. The topic is intriguing and interesting. To date, Prader-Willi syndrome represents a very complex and underrated genetic disorder. The manuscript is well-written and easily readable, despite some misspelled words (also in the title). However, it could be of interest to improve the quality of the manuscript with the information from this recent narrative review (PMID: 33671467).
Author Response
Thank you for your comments and have expanded the description of genes (imprinted and non-imprinted) in the 15q11-q13 region and have incorporated the new reference that you suggest (i.e., PMID: 33671467). This section has significantly been expanded and changes throughout the manuscript are in red.
Round 2
Reviewer 3 Report
no